# Uncommon Site of Metastasis and Prolonged Survival in Patients with Anaplastic Thyroid Carcinoma: A Systematic Review of the Literature

**DOI:** 10.3390/cancers12092585

**Published:** 2020-09-10

**Authors:** Aurora Mirabile, Matteo Biafora, Leone Giordano, Gianluigi Arrigoni, Maria Giulia Cangi, Italo Dell’Oca, Francesca Lira Luce, Davide Di Santo, Andrea Galli, Michele Tulli, Renata Mellone, Davide Valsecchi, Vanesa Gregorc, Mario Bussi

**Affiliations:** 1Department of Medical Oncology, IRCCS San Raffaele Scientific Institute, 20132 Milan, Italy; mirabile.aurora@hsr.it (A.M.); gregorc.vanesa@hsr.it (V.G.); 2Department of Otorhinolaryngology, IRCCS San Raffaele Scientific Institute, 20132 Milan, Italy; biafora.matteo@hsr.it (M.B.); liraluce.francesca@hsr.it (F.L.L.); disanto.davide@hsr.it (D.D.S.); galli.andrea@hsr.it (A.G.); tulli.michele@hsr.it (M.T.); bussi.mario@hsr.it (M.B.); 3Unit of Pathology, IRCCS San Raffaele Scientific Institute, 20132 Milan, Italy; arrigoni.gianluigi@hsr.it (G.A.); cangi.mariagiulia@hsr.it (M.G.C.); 4Radiotherapy Department, IRCCS San Raffaele Scientific Institute, 20132 Milan, Italy; delloca.italo@hsr.it; 5Department of Radiology, IRCCS San Raffaele Scientific Institute, 20132 Milan, Italy; mellone.renata@hsr.it; 6Emergency Department, IRCCS San Raffaele Scientific Institute, 20132 Milan, Italy; valsecchi.davide@hsr.it

**Keywords:** anaplastic thyroid carcinoma, subcutaneous metastasis, long survival, case report, infrequent metastatic sites

## Abstract

**Simple Summary:**

The therapeutic strategies employed for anaplastic thyroid cancer patients seems to be insufficient to prolong their survival, but some characteristics could predict a good prognosis, so that, starting from our experience we offer a systematic review of the literature to better understand anaplastic thyroid cancers behavior and their prognostic factors, in order to recognize and select the patients with the higher probability of better outcome even if metastatic. Moreover, we described an uncommon site of metastasis in order to improve scientific knowledge about this rare and highly aggressive pathology.

**Abstract:**

Anaplastic thyroid carcinoma (ATC) is a very rare, highly aggressive malignant thyroid tumor with an overall survival from 3 to 5 months in most of the cases. Even the modern and intensive treatments seem not to be enough to provide a cure, also for the resectable ones, and the role of chemotherapy is still unclear but does not seem to prolong survival. Nevertheless, some patients survive longer and have a better outcome, even in the presence of metastasis, than what the literature reports. We present the case of a 64-year-old female affected by ATC, treated on February 2018 with surgery followed by chemoradiation. One year after surgery, the patient developed a subcutaneous recurrence that was radically resected and is still alive 29 months after the diagnosis. We propose a systematic review of the literature to deepen the knowledge of the prognostic factors of ATC with the aim to recognize and select the patients with a better outcome, even if metastatic, and to describe a very uncommon site of metastatization.

## 1. Introduction

Anaplastic thyroid carcinoma (ATC) is a rare neoplasm, representing about 1–2% of all thyroidal malignancies, with a mean age at diagnosis between 55 and 65 years [1,2,3]; however, it contributes up to 14–50% of thyroid cancer-related deaths.

About 25% of ATC patients had a thyroid goiter and 10% had a family history of goiter. In fact, it is more common in places with endemic goiter due to iodine deficiency. Most ATC patients present with a thyroid mass (77%) followed by a node mass (54%), dysphagia (40%) and neck pain (26%).

It is a very aggressive malignancy that arises from follicular cells of the thyroid gland, with rapid growth and radio- and chemo-resistance; yet, even if it leads to a median survival of 3–10 months [4], some patients reach a longer survival, demonstrating that little is still known about these tumors. In the last years, several prognostic factors have been studied: most of them were clinical factors, and very recently genetic and molecular factors.

Because of the aggressive behavior of the ATC, between 20% and 50% of cases present distant metastasis at the time of diagnosis [5]: the most common sites are the lungs (80%), followed by bones (6–16%) and brain (5–13%) [6,7], while cutaneous and subcutaneous metastasis are rare and usually occur in the setting of disseminated neoplastic disease [8].

The aim of our manuscript is to review the literature about uncommon sites of metastasis and prolonged survival in patients affected by ATC. We start with the case of a 64-year-old woman with an unusual presentation of temporal subcutaneous single metastasis of ATC, 11 months after a total thyroidectomy for the primary tumor, and who has a global overall survival of 29 months at the time of this review.

## 2. Methods

In order to identify all potentially relevant scientific papers published until July 2020 and reporting original research on the survival and uncommon sites of metastasis of ATC, a systematic search of the Medline databases (https://www.pubmed.ncb.nlm.nih.gov/) was carried based on the following criteria:Full text papers;Available abstracts;Papers written in English;Clinical studies, reviews and case reports related to a prolonged survival compared to the literature data;Clinical studies, reviews and case reports related to uncommon sites of metastasis compared to the literature data;Clinical studies, reviews and case reports on survival prognostic and predictive factors.
Clinical studies were excluded if they met at least one of the following criteria:Pediatric or pregnancy patients;Interim reports;About differentiated thyroid carcinomas;Non-anaplastic thyroid carcinomas;Molecular studies;Preclinical studies;Non-English written language;No full text;Non uncommon site of metastasis;Management of ATC.

The following search strings, sorted by “best match”, were used in Medline: “long survival and anaplastic thyroid carcinoma”, “infrequent or uncommon site of metastasis and anaplastic thyroid carcinoma”, “subcutaneous metastasis and anaplastic thyroid carcinoma”, “skin metastasis and anaplastic Thyroid carcinoma”, “rare manifestation of anaplastic Thyroid carcinoma”.

Two authors (AM and MB) independently selected the articles according to the inclusion and exclusion criteria. Any disagreements or differences in selection of the eligible articles were resolved by consultation and discussion with a third assessor (LG). The date last searched was 16 July 2020. The PRISMA (Preferred Reporting Items for Systematic Reviews and Meta-Analyses) flow diagram (http://prisma-statement.org/PRISMAStatement/FlowDiagram.aspx) (Figure 1) was followed, according to the specific guidelines [9].

### Case Description

We present the case concerning a 64-year-old female patient without any comorbidities (also a non-smoker) who underwent a total thyroidectomy for a rapid growth of a multinodular goiter in February 2018. The pre-operative ultrasound showed the presence of a non-homogeneous lesion of 39 mm × 35 mm × 50 mm occupying the isthmic region and the lower portions of the thyroid lobes. At presentation, the patient’s blood examinations were normal, with blood cell counts < 10.0 × 10(9)/L and blood platelet counts < 300.0 × 10(9)/L.

On the definitive histological examination, we found the presence of a 50% MIB1 ATC of the epithelial type (WHO 2017 Classification of tumor of endocrine organs, IARC). It is composed of squamoid cohesive tumor nests, admixed with lymphocyte and granulocyte infiltration and residual areas of follicular carcinoma, widely invasive, extracapsular, diffusely infiltrating both the thyroid lobes and the perithyroidal soft tissues with multiple areas of vascular invasion and extended to the right cricothyroideal corner (Figure 2A,B). Stage: pT4apN0.

Next-generation sequencing (NGS) was performed for molecular characterization of the neoplasia using the Oncomine Comprehensive Assay v.3M (OCAv3, ThermoFisher Scientific, Waltham, MA, USA), according to manufacturer’s protocols. A mutation in the *NRAS* gene was identified (p.Gln61Arg); no mutations were found in the *BRAF* (v-raf murine sarcoma viral oncogene homolog B1), *KRAS* (Kirsten rat sarcoma viral oncogene homolog), *HRAS* (Harvey Rat Sarcoma Viral Oncogene Homolog), and *TP53* (tumor protein 53) genes.

On the FDG PET/CT (18-fluorodeoxyglucose positron emission tomography) check of June 2018 there was an evidence of residual disease in the right thyroid space and therefore the patients underwent adjuvant concurrent chemo-radiotherapy with three courses of high dose CDDP. Radiotherapy was delivered by means of Helical Tomotherapy. The volumes of the treatment were based on a planning PET/CT. The dose delivered was 66 Gy and 54 Gy on high risk and low risk volumes, respectively, in 30 fractions by the Simultaneous Integrated Boost (SIB) technique, obtaining a complete remission for 8 months.

In April 2019, the patient reported the appearance of a painless swelling in the right temporal region and so she underwent an FDG PET/CT (June 2019): a 12 mm right temporal subcutaneous lesion of non-unequivocal meaning, suspect for a neoplastic disease (Figure 3A); and a maxillo-facial MRI with contrast (3 June 2019): alteration in the context of the right temporal muscle of 13 mm × 10 mm × 6 mm with contrast-enhancement, suspected to be a neoplastic lesion (Figure 3B).

The lesion was then surgically removed under general anesthesia with sacrifice of the skin above the lesion and the temporal muscle fibers close to the lesion.

Final histological examination: subcutaneous metastasis of anaplastic thyroid cancer with metaplastic squamous areas, cancer-free resection margins, and mutational status: *BRAF*, *KRAS*, *HRAS* and *TP53* wild type, as well as the *NRAS codon 61* mutated. (Figure 2C,D).

At the MRI check of 26 August 2019 and FDG PET/CT of 29 August 2019, there is no evidence of disease as in the last MRI of 10 April 20 and the FDG PET/CT of 20 April 2020 (Figure 3C,D).

At the time of this paper, the patient is in good health, currently free of disease and continues the be followed-up.

## 3. Results

### 3.1. Prognostic Factors

On July 2020, the systematic review of the literature from the Medline database yielded 174 records. One-hundred and ten were excluded after reviewing the title and abstract. Respectively, a total of 42 and 28 articles were selected for full-text review and closer inspection to determine whether they met the eligibility criteria for clinical trials about overall ATC patients’ survival and uncommon sites of metastasis. 

Seventeen full-text articles screened for survival were excluded, the major reasons being (1) basket or histology-independent trials (*n* = 9, 53%); (2) management of other thyroid malignancies (*n* = 5, 29.4%); and (3) an aim to underline thyroid toxicities (*n* = 3, 17.6%) (Prisma 2009 Flow Diagram). 

Twenty-three papers on prolonged survival of ATC met the inclusion criteria and are listed in Table 1.

In 2001, Sugitani et al. [34] tried to identify ATC prognostic factors in order to individualize the proper therapeutic strategy. Studying 47 patients with ATC retrospectively (since 1976 to 1999), they devised a novel prognostic index (PI) based on the following four independent, unfavorable prognostic factors: (1) acute symptoms (duration of severe complaints, such as dysphonia, dysphagia, dyspnea, hoarseness and rapid tumor growth of <1 month); (2) leukocytosis (leukocyte count >10,000/mm^3^); (3) tumor size >5 cm; and (4) distant metastasis. Patients were assigned a PI score of 1 to 4. Patients with a PI = 1 experienced a 62% survival rate at 6 months, whereas no patients with a PI = 3 survived longer than 6 months and all patients with a PI = 4 died within 3 months.

This score was confirmed also in 2012 and 2018 [35,36] with the addition of advanced age (>60–70 years), male gender, extra thyroidal invasion, giant cell and pleomorphic pattern, while tumors with co-existing well-differentiated papillary thyroid carcinoma displayed a better outcome [35,37,38,39].

More recently, in 2019, ATC incidence and mortality were assessed via a join point regression analysis of 567 ATC patients selected from the Surveillance, Epidemiology, and End Results 18 Registries Research database. Two validated nomograms were made, based on two predictive models: Nomogram 1 considered age, tumor size and metastasis (all before surgery); and Nomogram 2 considered age, tumor size, metastasis, surgery and extrathyroidal extension (all after surgery). The first one is useful in preoperative prediction of survival time. The second one provides additional outcome-related information [40], in order to optimize multimodal interventions such as surgery, chemotherapy and radiation treatment, and to improve outcomes in case of both locoregional and metastatic disease, as well as to increase the quality of life and to reduce disease-related mortality [20].

Surgery in ATC is recommended when feasible, because it prolongs OS and its feasibility depends on the possibility to obtain a complete resection (R0) according to the American Thyroid Association (ATA) ATC treatment guidelines, where, in absence of an extrathyroidal extension, a total lobectomy or total thyroidectomy with lateral lymph node dissection is recommended, while, in the presence of an extrathyroidal extension, if an R1 resection is possible, en bloc resection is recommended [41].

Similar results were reported in a case load published by Yau et al. [24] on 50 patients in which the following variables were significantly associated with prolonged survival: radical surgery, age <65 years, absence of distant metastases and a lesion size <6 cm. Nevertheless, there is a wide consensus that surgery alone, even if curative, in most cases needs radiotherapy or concomitant radiotherapy and chemotherapy to achieve the best local control and survival outcomes [19,42,43].

In 2004, De Crevoisier et al. [25] reported the results of a multimodality prospective trial on 30 ATC patients and confirmed that a better long-term survival can be obtained by surgery followed by concomitant chemo radiation, underlining that more than half of the deaths were due to distant metastases rather than to local progression (68% vs. 5%), while in almost a third (27%) of cases it was related to the failure in both sites. 

The experience of Swaak-Kragten [26] confirmed the above data, showing higher overall survival (>5 years) and local control (89% complete remission) in patients undergoing initial surgery (R0/R1 resection) followed by concomitant chemoradiation with a total radiation dose >40 Gy (*p* < 0.001). This observation has been further underlined in a retrospective review in 2011 by Sherman et al. [27], where radiation dose (<60 Gy versus >60 Gy) and age (<70 years versus >70 years) were significant prognostic variables on both local control and overall survival in the multivariate analysis. 

In addition, the stage seems to have a prognostic importance. In 2016, Liu et al. published a clinical study of 50 cases of ATC, underling the 2-year overall survival of 40.0%, 31.0% and 6.3% for stage IVA, IVB and IVC, respectively (*p* < 0.05), which improved to 50% for stage IVA and IVB, if treated with combined surgery and radiotherapy compared to 35.7% with surgery alone (*p* < 0.05) [28].

More recently, these results were confirmed in 2018 by the meta-analysis of Xia Q. et al. [44] on 1163 patients with resectable ATC, where the combination of surgery and radiotherapy significantly reduced the risk of death compared with surgery alone (HR = 0.51, 95% CI: 0.36–0.73, *p* = 0.0002). The pooled analyses questioned the influence of chemotherapy on the overall survival of ATC patients (HR = 0.63, 95% CI: 0.33–1.21, *p* = 0.17).

In 2019, a review of the 25 published studies again confirmed these data but indicated that early multidisciplinary approaches using extensive radical surgery, in combination with adjuvant radiotherapy concomitant to taxanes or cisplatin-based chemotherapy, provided the best chance of disease control [45], underlining that probably tumor hystopathology can also influence the outcome, showing how the presence of foci of differentiated thyroid cancer within the anaplastic tumor, the absence of nodal involvement and of neutrophilic or lymphocytic infiltration together with a median dose of radiation of 50 Gy or more (median overall survival 10.5 months), the use of any surgery (median overall survival 10.5 months as well), the presence of a pre-existing tumor, the epithelial growth and a squamous cell carcinoma component were also associated with better survival [22,45,46].

Ugurlu et al. [12] in their systematic review underlined that noninvasive/microinvasive anaplastic thyroid carcinoma may be a different disease entity than classical anaplastic thyroid carcinoma, with a more favorable prognosis and longer disease-free survival. A minimal tumor capsular invasion [12] or an encapsulated tumor [13,14] could have an excellent effect on survival, regardless of age.

Recently, the understanding of the molecular aberrations in thyroid cancer has advanced the field, leading to one of the most important treatment discoveries: *BRAF V600E* was found as the most frequently mutated gene in thyroid cancer, ranging from 27.3% to 87.1% in papillary thyroid carcinoma (PTC) [47,48] but only 25–45% in ATC [48,49,50,51,52], followed by *NRAS*, *HRAS* or *KRAS* mutations (mutually exclusive with *BRAF*) that occurs, respectively, in 28% and 24% of PTCs and ATC [49], while *TP53* inactivation, highly prevalent in ATCs, and relatively rare in PDTCs (73% vs. 8%) [49], has been considered a hallmark of advanced thyroid tumors since ATC patients with *TP53* mutations were shown to have a trend towards a shorter time to failure after primary treatment [37,50].

Further regarding metastatic diseases, from our literature revision emerged only retrospective data of 779 patients [10,11,12,22,23,24,29,30,31,32,33,51,52,53], treated with a multimodality approach (Table 1) and with a median overall survival that ranges from 3 to 28 months.

In particular, Baek et al. in 2017 retrospectively studied 329 patients diagnosed at 19 medical centers in Korea and who were affected by local, locoregional advanced or metastatic ATC. Evaluating the survival outcomes according to the different clinical features and treatments, they identified age ≥70 years old, the presence of initial clinical symptoms, distant metastasis and treatment modality as significant risk factors (*p* < 0.05). Overall, the patients who underwent curative surgery and adjuvant radiotherapy or concurrent chemoradiation obtained the best survival (*p* < 0.05) [33].

In 2013, another retrospective study published by Sun et al. suggested the value of blood counts as an independent prognostic factor at the time of diagnosis. Analyzing the outcome of 60 patients over a 30-year period, it was concluded that the best prognosis was seen in younger patients (<55 years), without distant metastases, with white blood cell counts <10.0 × 10 (9)/L or blood platelet counts <300.0 × 10 (9)/L, treated with radiotherapy doses ≥40 Gy or surgery plus postoperative radiotherapy but without chemotherapy [30].

Lastly, Busnardo et al., in 2000, investigating the role of multimodality treatment (surgery, chemotherapy and radiotherapy) in 39 ATC patients, 26 of which had local disease, 13 metastatic at diagnosis (mainly to the lung) and 9 who had developed metastasis during the follow-up period, suggest that aggressive and appropriate combinations of treatment provide some survival benefit in patients with ATC. Preoperative chemotherapy and radiotherapy may enhance surgical resectability. Nevertheless, only a few patients responded to chemotherapy, so early detection is one of the most important prognostic factors because it leads to a curative approach, avoiding the need for chemotherapy since ATC is often resistant to anticancer drugs [18].

### 3.2. Unusual Metastatic Sites

Of the twenty-eight full-text articles screened for uncommon sites of metastasis, 23 were excluded due to the following reasons: (1) management of other thyroid malignancies (*n* = 8; 28.5%); (2) about the diagnostic and staging modalities (*n* = 7; 25%); (3) about the technicity of the medical/surgical approach (*n* = 7; 25%); (4) aim to underline thyroid toxicities (*n* = 4; 14.5%); and (5) basket or histology-independent trials (*n* = 2; 7%). In turn, five papers about uncommon metastatization met the inclusion criteria and are listed in Table 2.

Lungs are known as the most common metastatic sites, followed by the intrathoracic and neck lymph nodes, but starting from our experience of a singular subcutaneous metastasis, we systematically reviewed the literature to discover other less common sites.

Eighty percent of the uncommon cases evidenced by the systematic review were in younger (≤65 y) patients, mainly skin metastasis, and just one gastric metastasis, all leading to a poor prognosis shortly after the appearance of skin lesions, as described in Table 2.

Unfortunately, most of our knowledge derived from single-institution studies with a small number of patients and a short-term follow-up, or from case reports.

## 4. Discussion

Data from the literature demonstrated that ATC is a very rare, highly aggressive malignant thyroid tumor. The prognosis is poor, with a mortality rate of 68.4% at 6 months and 80.7% at 12 months, and a reported overall survival from 3 to 5 months, with a 1-year survival of approximately 20%, and a few patients surviving longer (<5%) despite the modern intensive treatment [58,59].

In local or locoregionally advanced disease, the median survival ranges from 1 to 240 months, with a median of 10 months and an average of 32 months, especially in young patients (<70 years) treated with radical surgery followed by radiotherapy with at least a 60 Gy radiation dose. Other favorable prognostic factors are the absence of signs or symptoms, without neutrophil or lymphocytes infiltration, without leukocytosis or thrombocytosis and distant metastasis. Also favorable are a slow tumor growth and tumor size <5 cm, female gender, absence of extrathyroidal invasion and nodal involvement, encapsulated tumor with no or minimal capsular invasion and the presence of areas of differentiated carcinomas without *BRAF* or *RAS* or *TP53* mutations.

The patient in the present case report presented several positive prognostic factors that may explain her long-term survival (Table 3). She underwent radical surgery followed by aggressive multimodality treatment with a 66 Gy radiation dose, concomitant to three courses of high-dose cisplatin (100 mg/m^2^ per course). She also had a tumor size less than 6 cm, the absence of distant metastases and no symptoms at presentation, the absence of leukocytosis or thrombocytosis at diagnosis, together with being <70 years old, as well as with the absence of nodal involvement and with areas of follicular carcinoma.

Nevertheless, she was affected by a widely invasive ATC, diffusely infiltrating both the thyroid lobes and the perithyroidal soft tissues with multiple areas of vascular invasion and the extension to the right cricothyroideus corner, with a rapid tumor grow and the presence of neutrophilic and lymphocytic infiltration as negative prognostic factors.

At less than one year after radical surgery she developed a single uncommon subcutaneous metastasis that usually leads to a rapid and negative evolution.

On the other side, she is alive and in a good heath more than 29 months after surgery and without evidence of disease more than 15 months after metastatic diagnosis, doubling the median survival by time from the systematic review of the long survival data and tripling the literature data.

Using the validated nomograms published by Qiu et al. [20], our patient reached 9.2 points on a post-operative assessment. According to Nomogram 2, her 1-, 3- and 5-year survival rates were approximately 45%, 30% and 25%, respectively, before metastatization (age 64 years = 5.5 points, extrathyroidal extension = 1.7 points and tumor diameter 5 cm = 2 points), but the score increased to 13.2 after subcutaneous metastasis onset (=4 points). Thus, the survival rate declined approximately to 18%, 9% and 7%, leading to the hypothesis that other factors could be involved in our patient’s favorable prognosis, since actually she is over 2.5 years of follow-up.

A reason of this good outcome could be searched probably in the tumor molecular characteristics, since it was the *BRAF*, *KRAS*, *HRAS* and *TP53* wild type that could at least partially contribute to the good outcome, but codon 61 of the *NRAS* gene was mutated, possibly worsening the outcome.

In fact, despite the lack of direct association between the *RAS* mutations and ATC-specific characteristics, both the univariate and multivariate analysis demonstrated a shorter OS in patients with a *RAS* mutation (*p* = 0.004), which identified this mutation as the only independent prognostic factor of poorly differentiated carcinomas in this series [60].

We set out to highlight the present case since, to our knowledge, this is the only case of a single subcutaneous metastasis. We found five case reports in the literature concerning different uncommon metastatization (Table 2), all related to a worse and rapid prognosis (within one month), but none similar to our presentation that seems to not compromise our patient’s outcome.

## 5. Conclusions

In spite of the heterogeneity of most retrospective experiences and case reports, our experience gave the chance to underline the importance of prognostic factors and the need for prospective studies to investigate new factors to better predict outcome, in order to recognize the potential responders even among patients with a highly aggressive and rare tumor such as ATC.

## Figures and Tables

**Figure 1 cancers-12-02585-f001:**
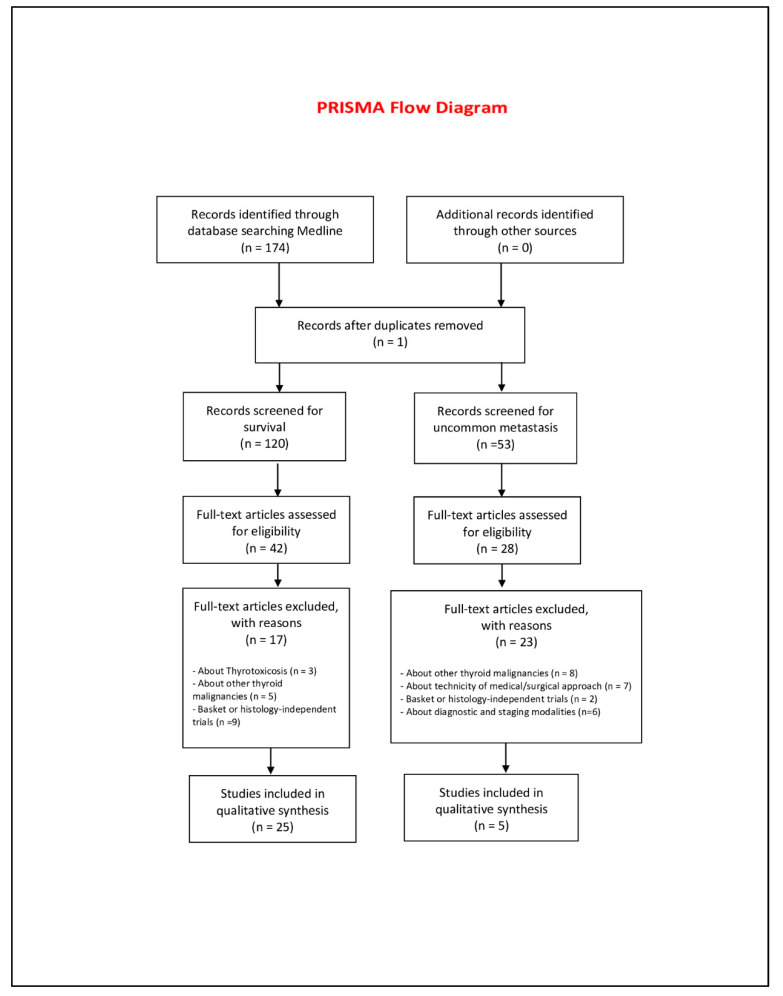
Prisma 2009 flow diagram [9].

**Figure 2 cancers-12-02585-f002:**
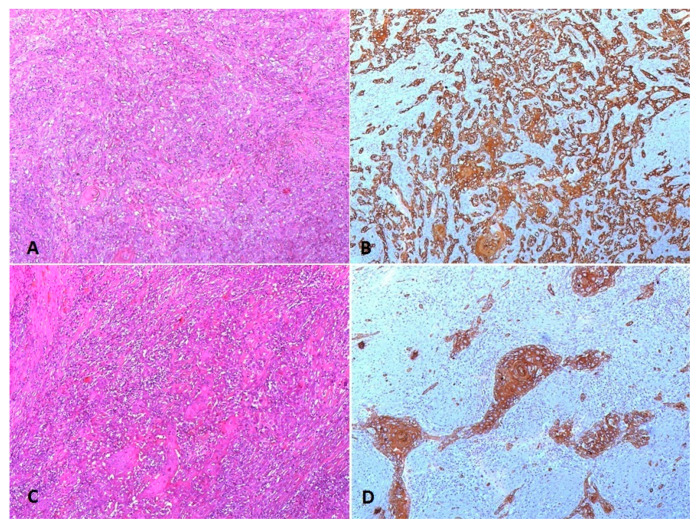
(**A**) Thyroid neoplastic cells of the anaplastic epithelial type with focal squamoid nests and admixed with lymphocytic infiltrate (HE 100×); (**B**) *CKAE1AE3* immunohistochemistry highlights the epithelial pattern of the neoplastic cells with a focal horn pearl (100×); (**C**) subcutaneous metastasis showing the prevalent squamoid pattern of the neoplastic epithelial cells (HE 100×); (**D**) *CKAE1AE3* immunohistochemistry highlights the neoplastic squamoid cells with focal horn pearls (100×).

**Figure 3 cancers-12-02585-f003:**
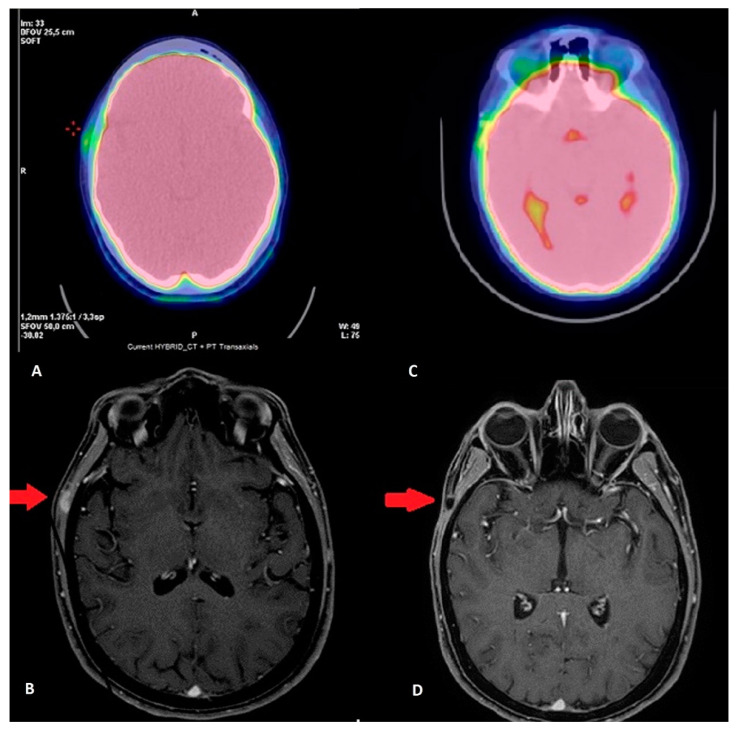
(**A**,**B**) Pre-operative imaging: (**A**) FDG PET/CT: shows the uptake of the lesion (green area in the upper-left part of the panel); (**B**) MRI: the red arrow indicates the lesion. (**C**,**D**) Twelve months after surgery: (**C**) FDG PET/CT shows no uptake; (**D**) MRI shows the surgical alteration of the temporal muscle without contrast-enhancement, and the red-arrow indicates the area.

**Table 1 cancers-12-02585-t001:** Selection of articles regarding the long survival of ATC patients.

Articles	Stage	Treatment	Evidence	Risk Factors	Median Survival
Stenman (2020) [10]	Locally advanced	CT/RT->CH	Case Report	BRAF neg, P53 pos, PDL1 80%, Ki67 30%	12 mos
Kanazawa (2019) [11]	Locally advanced	CH->RT and Radioactive Iodine	Case Report	Ki 67 < 30%, PI = 0	84 mos
Uğurlu (2018) [12]	Early	CH->RT	Case Report	No (encapsulated and early stage)	72 mos
Ito (2003) [13]	T4aN0	CH	Case Report	No (encapsulated)	57 mos
Dibelius (2014) [14]	Early	CH	Case Report	No (Not invasive)	14 mos
Lowe (2014) [15]	Local/locoregional and M+	Multimodality	Retrospective (20 pts)	Various	6 mos (range 1–11 mos)
Siironen (2010) [16]	Local/locoregional and M+	Multimodality	Retrospective (44 pts)	Various	11.6 mos
Kurukahvecioglu (2007) [17]	Locally advanced	CH	Case Report	P53 and Ki 67 pos, soft tissue involvement	36 mos
Busnardo (2000) [18]	Local/locoregional and M+	Multimodality	Retrospective (39 pts)	Various	5.7 mos
Derbel (2011) [19]	Local/locoregional and M+	Multimodality	Retrospective (44 pts)	Various	8 mos
Haddad (2005) [20]	Locally advanced	CH->CT/RT	Case Report	Extracapsular, positive margins	24 mos
Liu (2006) [21]	Locally advanced	CH	Case Report	Extracapsular	240 mos
Dumke (2014) [22]	Local or locoregional	Multimodality	Retrospective (40 pts)	Various	10.5 mos
Guimaraes (2000) [23]	Early	CH	Case Report	Extracapsular foci of ATC	35 mos
Sugitani (2001–2012–2018) [10,11,12]	Local/locoregional and M+	Multimodality	Retrospective (47 pts)	Positive/Negative risk factors	>6 mos/<6 mos
Yau (2008) [24]	Locally advanced	CH-> RT	Retrospective (50 pts)	<65 y, N0, M0, papillary carcinoma	3 mos (rage 4 days–16 years)
De Crevoisier (2004) [25]	Locoregional and M+	Multimodality	Retrospective (30 pts)	Various	10 mos
Swaak-Kragten (2009) [26]	Local/locoregional and M+	Multimodality	Retrospective (75 pts)	Various	2.9 mos (0–119 mos)
Sherman (2011) [27]	Local/locoregional	Multimodality	Retrospective (37 pts)	Various	6 mos
Liu (2016) [28]	Local/locoregional and M+	Multimodality	Retrospective (50 pts)	Various	24 mos (range 24–48 mos)
Brignardello (2014) [29]	Local/locoregional	Multimodality	Retrospective (55 pts)	Various	6.57 mos
Sun (2013) [30]	Local/locoregional and M+	Multimodality	Retrospective (60 pts)	Various	8 mos
Palestini (2010) [31]	Local/locoregional and M+	Multimodality	Retrospective (20 pts)	Various	8 mos (range 3–28 mos)
Roche (2010) [32]	Local/locoregional and M+	Multimodality	Retrospective (26 pts)	Various	4 mos
Baek (2017) [33]	Local/locoregional and M+	Multimodality	Retrospective (329 pts)	Various	8 mos

pts = patients; mos = months; multimodality = surgery alone (S) or followed by radiotherapy (RT) or followed by chemoradiation (CT/RT).

**Table 2 cancers-12-02585-t002:** Selection of articles regarding uncommon sites of ATC metastasis.

Articles	Age	Stage	M+	Treatment	Evidence	Risk Factors	Survival
Danialan (2016) [54]	65 y	Locally advanced	Lung skin	CH->RT/CT+ Pazopanib+ target therapy	Case Report	BRAF pos	9 mos
Altinay (2014) [53]	57 y	Advanced	Subcarinal, paratracheal, aortopulmonary, trachea-bronchial and mediastinal->skin	CT	Case Report	ND	1 mo
Ayaz (2015) [55]	72 y	Advanced	Lung skin gastric	/	Case Report	ND	1 mo
Lim (2010) [56]	63 y	Advanced	Skin	CT/RT	Case Report	ND	2 mos
Hassan (2017) [57]	62 y	Advanced	Soft tissue	/	Case Report	ND	1 mo

M+ indicates sites of metastasis.

**Table 3 cancers-12-02585-t003:** Prognostic factors from the literature data and in our patient.

Prognostic Factors		
Favorable	Our Patient’s Prognostic Factors	References
Coexisting well differentiated carcinoma	X	Salehian (2019) [45] Hirokawa (2016) [46] Dumke (2014) [22] Sugitani (2012) [35] Rao (2017) [37] Kebebew (2005) [38] Kim (2007) [39]
Radical surgery	X	Baek (2017) [33] Yau (2008) [24]
Tumor size < 6 cm	X	Yau (2008) [24]
Age < 70aa	X	Baek (2017) [33] Sherman (2011) [27]
Female gender	X	Sugitani (2012–2018) [35,36] Sherman (2011) [27]
Leucocyte blood count < 10 × 10^9^	X	Sun (2013) [30]
Platelet count < 300 ×10^9^	X	Sun (2013) [30]
No nodal involvement	X	Sugitani (2012–2018) [35,36]
Squamous cell carcinoma components		Salehian (2019) [45] Hirokawa (2016) [46] Dumke (2014) [22]
Concomitant chemo-radiotherapy	X	Baek (2017) [33] Kim (1987) [42] Perri (2011) [43] Derbel (2011) [19] De Crevoiser (2004) [25] Swaak-Kragten (2009) [26] Liu (2016) [28] Xia (2018) [44] Salehian (2019) [45]
Radiotherapy > 60 Gy	X	Sherman (2011) [27]
**Unfavorable**		
Acute symptoms		Sugitani (2001) [34]
Rapid tumor growth	X	Sugitani (2012) [35] Rao (2017) [37] Kebebew (2005) [38] Kim (2007) [39]
Distant metastasis	X	Sugitani (2001) [34] Baek (2017) [33]
Age > 60 y.o.	X	Baek (2017) [33]
Extrathyroidal invasion	X	Baek (2017) [33]
Giant cell and pleomorphic pattern		Baek (2017) [33]
Capsular invasion	X	Uğurlu,(2018) [12]
Lymphocytic and neutrophilic infiltration	X	Salehian (2019) [45] Hirokawa (2016) [46] Dumke (2014) [22]
*TP*53 mutation		Rao (2017) [37] Ubertini (2015) [50]
*RAS* mutation	X	Volante (2009) [60] Landa (2016) [37]
*BRAF* mutation		Landa (2016) [49]

X: present.

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
