# Peer review of "Uncommon Site of Metastasis and Prolonged Survival in Patients with Anaplastic Thyroid Carcinoma: A Systematic Review of the Literature"

_cancers, 2020, doi:10.3390/cancers12092585_

Round 1
Reviewer 1 Report
This work presents a case of anaplastic thyroid carcinoma (ATC) in a 64 y.o. female patient who received surgery and combination chemo- and radiotherapy, and then developed a subcutaneous metastasis in the right temporal area about a year after the primary treatment. The lesion was surgically removed, and the patients remains alive about 2.5 years after diagnosis.
The authors performed systematic review of the literature aimed at the identification of factors associated with better ATC outcome, and to define uncommon ATC metastatic sites. After that, the described case is put in context of findings from the literature.
ATC is a relatively rare yet extremely aggressive type of human cancer generally with fulminant progression, very short survival times and unfavorable prognosis. At the same time, a small proportion of patients survive for a noticeably longer time; the reasons are not fully understood and explained. It therefore is clinically important to continue gaining further insights into the complex of ATC particularities and treatment modalities to determine patients with better prognosis and specific treatment approaches from which ATC patients may benefit. In this regard, the work is a useful and timely report, well suitable for the special issue "Thyroid Cancer Metastases (Biological and Clinical Aspects)".
Several points need to be addressed.
- The Results section needs to be better structured. Please make subsections for i) prognostic factors of ATC and ii) unusual metastatic sites (despite the latter one is relatively short, lines 258-265). Many paragraphs seem choppy, contain just one sentence, making an impression of an unfinished idea. These need to be combined if they are united by a common thought.
- PRISMA flow diagram does not look fully compliant as it does not consistently present excluded records/articles in outside-directed boxes. In the right column, of 28 full-text articles assessed for eligibility 28 were excluded, yet 5 were included in the study – it is unclear how. What is “Thyroid Toxicities”?
- Major findings for prognostic factors should be summarized in a table (like Table 2 for metastatic sites). This will make the paper easier to read.
- First three paragraphs of the Discussion in the present form rather relate to the Results. The first paragraph lists epidemiological data on ATC but presented pieces of information seem to contradict each other. What does “mortality rate of >90%” mean (line 268)? What is time unit here? If ATC has the “reported overall survival from 3 to 5 months”, then how 1-year survival could be 20% and how these data correspond to 90% mortality rate? Please make these statements clear. The second paragraph (histopathology) seems to be contradicting to the information on lines 211-218; please rephrase or remove it. It would be easier to understand the message of these paragraphs if an introductory phrase is added before (e.g. “Data from the literature demonstrate that ATC is…”). After that, it is, again, an introductory sentence explaining that an attempt to project the knowledge of prognostic factors from the literature to the presented ATC case is made, would make the paper easier to follow. Further, it would be advisable to make separate subsection outlining favorable and unfavorable factors seen in the presented case.
- Lines 305-307. Is there any data in the literature suggesting that NRAS codon 61 mutation as compared to other major genetic alterations could be associated with better prognosis? Please include such references or state that it is unknown.
Specific comments
- Abstract, l.21. “to heal” – “to provide cure”?
- Abstract, sentence on l.22-23. Please change word order e.g. by placing “even in presence of metastasis” after “longer” or “outcome”.
- Methods, l.50. Please provide URL for “Medline databases”. Why plural is used?
- The sentence on l.76-79 does not flow well, please rephrase.
- Human gene names should be italicized throughout the text.
- L.105: “indolent” – “painless”?
- L.106: “performed “ – “underwent”?
- Pages 3-4. Please, be consistent to use “PET-CT” or, preferably “PET/CT” throughout the text.
- Please, revise Fig.2 legend to make easier to understand and avoid duplications. Include the tracer type; “captation” – “uptake”?
- L.131: “global” – “overall”?
- L.155: “a PI score of 1 to 46”? Maybe “4” instead of “46”?
- L.158: “addiction” – “addition”?
- L.173: please place “total” before “thyroidectomy”.
- L.188: please, insert “the” before “above”.
- L.190: this part of the sentence does not flow well, please rephrase.
- L.194-195: please rephrase “few patients underwent to radical surgery”.
- L. 205: please rephrase “the value of chemotherapy in prolong the survival…”.
- L.211-218. Please extensively review this paragraph. It is a very long single sentence which is difficult to follow. In addition, both the presence of differentiated foci on histology and the absence of nodal disease, etc. seem to be favorable factors. Why then they are opposed to each other through the “while”?
- L.222: please make clear which “capsular invasion” is meant – the tumor capsule or thyroid capsule.
- L.225-230. Again, this is a very long sentence difficult to follow. Please edit it carefully.
Author Response
Thank you for your interest in our manuscript and for your precious suggestions, which have greatly helped improve the manuscript.
- The Results section needs to be better structured. Please make subsections for i) prognostic factors of ATC and ii) unusual metastatic sites (despite the latter one is relatively short, lines 258-265). Many paragraphs seem choppy, contain just one sentence, making an impression of an unfinished idea. These need to be combined if they are united by a common thought.
Response: Thanks for your interest in our study. We modified the results as suggested.
- PRISMA flow diagram does not look fully compliant as it does not consistently present excluded records/articles in outside-directed boxes. In the right column, of 28 full-text articles assessed for eligibility 28 were excluded, yet 5 were included in the study – it is unclear how. What is “Thyroid Toxicities”?
Response: we have now corrected the diagram
- Major findings for prognostic factors should be summarized in a table (like Table 2 for metastatic sites). This will make the paper easier to read.
Response: We summarized prognostic factors in table 3 but it’s not possible to specify the survival benefits for each factor.
- First three paragraphs of the Discussion in the present form rather relate to the Results. The first paragraph lists epidemiological data on ATC but presented pieces of information seem to contradict each other. What does “mortality rate of >90%” mean (line 268)? What is time unit here? If ATC has the “reported overall survival from 3 to 5 months”, then how 1-year survival could be 20% and how these data correspond to 90% mortality rate? Please make these statements clear. The second paragraph (histopathology) seems to be contradicting to the information on lines 211-218; please rephrase or remove it. It would be easier to understand the message of these paragraphs if an introductory phrase is added before (e.g. “Data from the literature demonstrate that ATC is…”). After that, it is, again, an introductory sentence explaining that an attempt to project the knowledge of prognostic factors from the literature to the presented ATC case is made, would make the paper easier to follow. Further, it would be advisable to make separate subsection outlining favorable and unfavorable factors seen in the presented case.
Response: We now have modified as requested
- Lines 305-307. Is there any data in the literature suggesting that NRAS codon 61 mutation as compared to other major genetic alterations could be associated with better prognosis? Please include such references or state that it is unknown.
Response: We have now better explained this information and the references to the text.
- Abstract, l.21. “to heal” – “to provide cure”?
Response: changed as requested
- Abstract, sentence on l.22-23. Please change word order e.g. by placing “even in presence of metastasis” after “longer” or “outcome”.
Response: changed as requested
- Methods, l.50. Please provide URL for “Medline databases”. Why plural is used?
Response: added as requested
- The sentence on l.76-79 does not flow well, please rephrase.
Response: rephrased as requested
- Human gene names should be italicized throughout the text.
Response: changed as requested
- L.105: “indolent” – “painless”?
Response: changed as requested
- L.106: “performed “ – “underwent”?
Response: changed as requested
- Pages 3-4. Please, be consistent to use “PET-CT” or, preferably “PET/CT” throughout the text.
Response: changed as requested
- Please, revise Fig.2 legend to make easier to understand and avoid duplications. Include the tracer type; “captation” – “uptake”?
Response: changed as requested
- L.131: “global” – “overall”?
Response: changed as requested
- L.155: “a PI score of 1 to 46”? Maybe “4” instead of “46”?
Response: corrected as requested
- L.158: “addiction” – “addition”?
Response: corrected as requested
- L.173: please place “total” before “thyroidectomy”.
Response: changed as requested
- L.188: please, insert “the” before “above”.
Response: changed as requested
- L.190: this part of the sentence does not flow well, please rephrase.
Response: changed as requested
- L.194-195: please rephrase “few patients underwent to radical surgery”.
Response: changed as requested
- L. 205: please rephrase “the value of chemotherapy in prolong the survival…”.
Response: rephrased as requested
- L.211-218. Please extensively review this paragraph. It is a very long single sentence which is difficult to follow. In addition, both the presence of differentiated foci on histology and the absence of nodal disease, etc. seem to be favorable factors. Why then they are opposed to each other through the “while”?
Response: reviewed as requested
- L.222: please make clear which “capsular invasion” is meant – the tumor capsule or thyroid capsule.
Response: clarified as requested
- L.225-230. Again, this is a very long sentence difficult to follow. Please edit it carefully.
Response: edited as requested
Reviewer 2 Report
The manuscript entitled "Uncommon site of metastasis and prolonged survival in patients with anaplastic thyroid carcinoma: a systematic review of literature." is a very comprehensive review focusing the attention on the prognostic factors in anaplastic thyroid carcinoma.
Minor comments:
- The Introduction section is too short. The Authors should improve this section.
- In the Case description section, could the Authors better explain the assay adopted for the molecular analysis?
- Page 7 line 155 "Patients were assigned a PI score of 1 to 46.", the Authors should control the range for PI score.
- The Authors should provide the extensive form for all acronyms, including gene acronyms, through the text when they first appear.
- Gene acronyms should be written in italics.
Author Response
Thank you for your interest in our manuscript and for your precious suggestions, which have greatly helped improve the manuscript.
- The Introduction section is too short. The Authors should improve this section.
Response: improved as requested
- In the Case description section, could the Authors better explain the assay adopted for the molecular analysis?
Response: explained as requested
- Page 7 line 155 "Patients were assigned a PI score of 1 to 46.", the Authors should control the range for PI score.
Response: corrected as requested
- The Authors should provide the extensive form for all acronyms, including gene acronyms, through the text when they first appear.
Response: provided as requested
- Gene acronyms should be written in italics.
Response: modified as requested
Round 2
Reviewer 1 Report
The manuscript has been greatly improved. Some stylistic and language changes are still required.
- Abstract, sentence on L.24-26. Please split it into two putting a full stop after “chemoradiation” on L.25. Begin the next sentence with “One year after surgery, the patient developed a subcutaneous…”. It would be easier to read and follow.
- L.38: “lack” – “deficiency”.
- L.39: please delete “(40%)-dysphagia” and add a comma after (54%).
- L.41: “grow” – “growth”; also on L.91, Table 3, L.308.
- L.42: Combine this para with the previous one into one para.
- L.45-49: combine these two sentences into one para. Please add “the” before “lungs” on L.47.
- L.87: “have” – “has”.
- L.90: please begin the sentence with “We present the case of a 64-year-old…”. L.91: “rapid grow” – “rapid growth of”.
- L.96: “It’s composed by” – “It is composed of”.
- L97: please delete “s” at the end of “lymphocytes” and “granulocytes” to read “lymphocyte” and “granulocyte”, respectively.
- L.103: please insert “the” before NRAS.
- L.104: please insert “the” before BRAF.
- L.106: here and elsewhere (L.134, 238, 240, 294, 322, Table 3) please change “Tp53” to “TP53”, italicized.
- L.113: please begin the sentence with “On a PET/CT scan of June 2018, there was…”.
- L 114: please place “adjuvant” before “concurrent” and delete “treatment”.
- L.117: please insert commas before and after “respectively”.
- L.117. Please combine this sentence with the previous para. Instead of “subsequent” please indicate month/year or follow-up months.
- L.123: please combine this sentence with the previous para.
- L.127: does “upper right” mean “upper left” here? Please make sure what is correct. Please also indicate what kind of tracer was used for PET/CT (this was requested in the previous review!).
- L.132: please delete “reported”. Combine the three para on L.132-138 into one para. L.135: “PET” is “PET/CT”?; please change “in” for “on”; L.136: please delete “in the” before “PET/CT”.
- L.145: the number “33” does not appear in the PRISM diagram. Is this a mistype?
- L.150, the diagram in Fig.1 and then on L.269: the term “thyroid toxicities” is uncommon and perplexing. Please use internationally accepted wording. This was requested in the first review.
- Table 1: please change “CH” for “S” or something more suitable meaning “surgery alone”.
- L.162: please begin the sentence just with “In 2001,…”.
- L.172: please rephrase “appeared far better” as e.g. “displayed better score/prognosis/outcome”.
- L178: “first-one useful” – “first one is useful”; please put a full stop after “time”. Begin the next sentence with “The second one provides…”.
- L.180: please delete a comma after “chemotherapy” and add it after “treatment”.
- L.181: please replace “in addition” with e.g. “as well as”.
- L.182: please insert “it” between “because” and “prolongs”.
- L.182-187: please combine these two para into one.
- L.193: please begin the sentence as “In 2004,…”.
- L.196-198: please combine this para with the previous one.
- L.200: please delete “to”.
- L.201: “chemo radiation” – “chemoradiation”.
- L.202: please begin the sentence with “The total radiation dose…” and combine this para (L.202-206) with the previous one.
- L.211: “metanalysis” – “meta-analysis”.
- L.213 and L.215: Z-values are excessive here, suggest removing.
- L.216-220: please combine these two para in one.
- L.220: ”the hystopathology” – “”tumor histopathology”.
- L.222-226: the sentence is incomplete; it is expected to end with something like “was also associated with better survival”. Please revise.
- L.216-226: please combine in one para.
- L.227-230: please delete “Moreover”, “how probably” – “that”, “appears to” – “may”, “from – “than”. L.230: please delete “Surely”, L.231: please insert e.g. “effect on” after “excellent”.
- L. 232: “this disease” – “thyroid cancer’.
- L.235: “ATCs” – “ATC”; “direct” – “directed”.
- L.235-237. This sentence is logically imbalanced. The BRAF mutation is mentioned as a therapeutic target while other mutations as “mutually exclusive” with BRAF. It should be carefully revised.
- L.238, 240: “Tp53” – “TP53”.
- L.232-241: please combine in one para.
- L.242-244: this sentence does not flow well, please rephrase.
- L.251: please delete “C” after “Sun”.
- L.263: please insert “it” after “Because”.
51. L.266: please begin with “Of twenty-eight…”; there seems to be an omission of the number before “were”, perhaps “23”? - L.266-271: please combine in one para.
- L.275: “systematic” – “systematically”.
- L.277: please delete “The” at the beginning of the sentence, add “in” after “were” and change “young” for “younger”.
- L.278: please delete “made by”, “by” before “gastric”, and delete “but”.
- L. 279: “from” – “after”, and delete “the” before “skin”.
- L.280: “derives” – “derived”.
- L.283: “Date” – “Data”.
- L.286: please insert a comma after “20%”; “survive” – “surviving”.
- L.287: please insert a comma after “disease”.
- L.291: “Is also favorable…” – “Also favorable are…”.
- L.292: “extra thyroid” – “extrathyroidal”.
- L. 293: “minimal capsular invasion or an encapsulated tumor” – “encapsulated tumor with no or minimal capsular invasion”.
- L.299: “signs or” – “no”; please delete “the” at the end of the line.
- Table 3.
65.1. Since the favorable prognostic features in the patient are presented in the manuscript before the unfavorable ones, it is reasonable to present those in the table first.
65.2. It is unnecessary to include the authors initials (or given names) in the rightmost column; the last name, the year and the reference number are sufficient. “Mustafa Umit Ugurlu, (2018)” – “Ugurlu (2018).
65.3. Age “aa” – “y.o.”.
65.4. “Extra thyroidal” – “Extrathyroidal”.
65.5. “Tp53 invasion” – “TP53 mutation”.
65.6. “Capsular invasion”: there is no mentioning of tumor encapsulation in the pathological description on P.3. Please add this information somewhere on L.95-100.
65.7. “mutations” – “mutation”.
65.8. Radiotherapy “>60y” – “>60Gy”.
- L.309. Please delete “In fact, at” at the beginning of the sentence.
- L.312-313: “median survival emerged by” – “median survival time from”.
- L.314: please add a comma before “our” and change “in” to “on”.
- L.316: please add a comma after “respectively”; “extra thyroid” – “extrathyroidal”.
- L.317: “points” – “score’.
- L.318: “Thus survival rate reduced… - “Thus, survival rate declined…”.
- L.322: “correlate” – “contribute”.
- L.323: please insert a comma after “outcome”, delete “demonstrated” and change “leading to a worse outcome” to “possibly worsening the outcome”.
- L.328: “Moreover, we wanted highlight our case because to our knowledge…” – “We set out to highlight the presented case since, to our knowledge,…”.
- L.335: “potentially” – “potential”; L.336: please insert “patients with such a” after “among”.
Author Response
We answered to all the revisions and edited as requested:
- Abstract, sentence on L.24-26. Please split it into two putting a full stop after “chemoradiation” on L.25. Begin the next sentence with “One year after surgery, the patient developed a subcutaneous…”. It would be easier to read and follow.
- L.38: “lack” – “deficiency”.
- L.39: please delete “(40%)-dysphagia” and add a comma after (54%).
- L.41: “grow” – “growth”; also on L.91, Table 3, L.308.
- L.42: Combine this para with the previous one into one para.
- L.45-49: combine these two sentences into one para. Please add “the” before “lungs” on L.47.
- L.87: “have” – “has”.
- L.90: please begin the sentence with “We present the case of a 64-year-old…”. L.91: “rapid grow” – “rapid growth of”.
- L.96: “It’s composed by” – “It is composed of”.
- L97: please delete “s” at the end of “lymphocytes” and “granulocytes” to read “lymphocyte” and “granulocyte”, respectively.
- L.103: please insert “the” before NRAS.
- L.104: please insert “the” before BRAF.
- L.106: here and elsewhere (L.134, 238, 240, 294, 322, Table 3) please change “Tp53” to “TP53”, italicized.
- L.113: please begin the sentence with “On a PET/CT scan of June 2018, there was…”.
- L 114: please place “adjuvant” before “concurrent” and delete “treatment”.
- L.117: please insert commas before and after “respectively”.
- L.117. Please combine this sentence with the previous para. Instead of “subsequent” please indicate month/year or follow-up months.
- L.123: please combine this sentence with the previous para.
- L.127: does “upper right” mean “upper left” here? Please make sure what is correct. Please also indicate what kind of tracer was used for PET/CT (this was requested in the previous review!).
- L.132: please delete “reported”. Combine the three para on L.132-138 into one para. L.135: “PET” is “PET/CT”?; please change “in” for “on”; L.136: please delete “in the” before “PET/CT”.
- L.145: the number “33” does not appear in the PRISM diagram. Is this a mistype?
- L.150, the diagram in Fig.1 and then on L.269: the term “thyroid toxicities” is uncommon and perplexing. Please use internationally accepted wording. This was requested in the first review.
- Table 1: please change “CH” for “S” or something more suitable meaning “surgery alone”.
- L.162: please begin the sentence just with “In 2001,…”.
- L.172: please rephrase “appeared far better” as e.g. “displayed better score/prognosis/outcome”.
- L178: “first-one useful” – “first one is useful”; please put a full stop after “time”. Begin the next sentence with “The second one provides…”.
- L.180: please delete a comma after “chemotherapy” and add it after “treatment”.
- L.181: please replace “in addition” with e.g. “as well as”.
- L.182: please insert “it” between “because” and “prolongs”.
- L.182-187: please combine these two para into one.
- L.193: please begin the sentence as “In 2004,…”.
- L.196-198: please combine this para with the previous one.
- L.200: please delete “to”.
- L.201: “chemo radiation” – “chemoradiation”.
- L.202: please begin the sentence with “The total radiation dose…” and combine this para (L.202-206) with the previous one.
- L.211: “metanalysis” – “meta-analysis”.
- L.213 and L.215: Z-values are excessive here, suggest removing.
- L.216-220: please combine these two para in one.
- L.220: ”the hystopathology” – “”tumor histopathology”.
- L.222-226: the sentence is incomplete; it is expected to end with something like “was also associated with better survival”. Please revise.
- L.216-226: please combine in one para.
- L.227-230: please delete “Moreover”, “how probably” – “that”, “appears to” – “may”, “from – “than”. L.230: please delete “Surely”, L.231: please insert e.g. “effect on” after “excellent”.
- L. 232: “this disease” – “thyroid cancer’.
- L.235: “ATCs” – “ATC”; “direct” – “directed”.
- L.235-237. This sentence is logically imbalanced. The BRAF mutation is mentioned as a therapeutic target while other mutations as “mutually exclusive” with BRAF. It should be carefully revised.
- L.238, 240: “Tp53” – “TP53”.
- L.232-241: please combine in one para.
- L.242-244: this sentence does not flow well, please rephrase.
- L.251: please delete “C” after “Sun”.
- L.263: please insert “it” after “Because”.
51. L.266: please begin with “Of twenty-eight…”; there seems to be an omission of the number before “were”, perhaps “23”? - L.266-271: please combine in one para.
- L.275: “systematic” – “systematically”.
- L.277: please delete “The” at the beginning of the sentence, add “in” after “were” and change “young” for “younger”.
- L.278: please delete “made by”, “by” before “gastric”, and delete “but”.
- L. 279: “from” – “after”, and delete “the” before “skin”.
- L.280: “derives” – “derived”.
- L.283: “Date” – “Data”.
- L.286: please insert a comma after “20%”; “survive” – “surviving”.
- L.287: please insert a comma after “disease”.
- L.291: “Is also favorable…” – “Also favorable are…”.
- L.292: “extra thyroid” – “extrathyroidal”.
- L. 293: “minimal capsular invasion or an encapsulated tumor” – “encapsulated tumor with no or minimal capsular invasion”.
- L.299: “signs or” – “no”; please delete “the” at the end of the line.
- Table 3.
65.1. Since the favorable prognostic features in the patient are presented in the manuscript before the unfavorable ones, it is reasonable to present those in the table first.
65.2. It is unnecessary to include the authors initials (or given names) in the rightmost column; the last name, the year and the reference number are sufficient. “Mustafa Umit Ugurlu, (2018)” – “Ugurlu (2018).
65.3. Age “aa” – “y.o.”.
65.4. “Extra thyroidal” – “Extrathyroidal”.
65.5. “Tp53 invasion” – “TP53 mutation”.
65.6. “Capsular invasion”: there is no mentioning of tumor encapsulation in the pathological description on P.3. Please add this information somewhere on L.95-100.
65.7. “mutations” – “mutation”.
65.8. Radiotherapy “>60y” – “>60Gy”.
- L.309. Please delete “In fact, at” at the beginning of the sentence.
- L.312-313: “median survival emerged by” – “median survival time from”.
- L.314: please add a comma before “our” and change “in” to “on”.
- L.316: please add a comma after “respectively”; “extra thyroid” – “extrathyroidal”.
- L.317: “points” – “score’.
- L.318: “Thus survival rate reduced… - “Thus, survival rate declined…”.
- L.322: “correlate” – “contribute”.
- L.323: please insert a comma after “outcome”, delete “demonstrated” and change “leading to a worse outcome” to “possibly worsening the outcome”.
- L.328: “Moreover, we wanted highlight our case because to our knowledge…” – “We set out to highlight the presented case since, to our knowledge,…”.
- L.335: “potentially” – “potential”; L.336: please insert “patients with such a” after “among”.
Thank you again for your interest in our manuscript and for your very precious suggestions, which have greatly helped improve the manuscript.
This manuscript is a resubmission of an earlier submission. The following is a list of the peer review reports and author responses from that submission.